# Development of an Endoscopic Auto-Fluorescent Sensing Device to Aid in the Detection of Breast Cancer and Inform Photodynamic Therapy

**DOI:** 10.3390/metabo12111097

**Published:** 2022-11-11

**Authors:** Brandon Gaitan, Collin Inglut, Udayakumar Kanniyappan, He N. Xu, Emily F. Conant, Lucas Frankle, Lin Z. Li, Yu Chen, Huang-Chiao Huang

**Affiliations:** 1Fischell Department of Bioengineering, University of Maryland, College Park, MD 20742, USA; 2Department of Radiology, University of Pennsylvania, Philadelphia, PA 19104, USA; 3Department of Biomedical Engineering, University of Massachusetts, Amherst, MA 01003, USA

**Keywords:** optical redox ratio, breast cancer, autofluorescence, molecular endoscopic imaging, photodynamic therapy

## Abstract

Breast cancer is the most diagnosed cancer type in women, with it being the second most deadly cancer in terms of total yearly mortality. Due to the prevalence of this disease, better methods are needed for both detection and treatment. Reduced nicotinamide adenine dinucleotide (NADH) and flavin adenine dinucleotide (FAD) are autofluorescent biomarkers that lend insight into cell and tissue metabolism. As such, we developed an endoscopic device to measure these metabolites in tissue to differentiate between malignant tumors and normal tissue. We performed initial validations in liquid phantoms as well as compared to a previously validated redox imaging system. We also imaged ex vivo tissue samples after modulation with carbonyl cyanide 4-(trifluoromethoxy) phenylhydrazone (FCCP) and a combination of rotenone and antimycin A. We then imaged the rim and the core of MDA-MB-231 breast cancer tumors, with our results showing that the core of a cancerous lesion has a significantly higher optical redox ratio ([FAD]/([FAD] + [NADH])) than the rim, which agrees with previously published results. The mouse muscle tissues exhibited a significantly lower FAD, higher NADH, and lower redox ratio compared to the tumor core or rim. We also used the endoscope to measure NADH and FAD after photodynamic therapy treatment, a light-activated treatment methodology. Our results found that the NADH signal increases in the malignancy rim and core, while the core of cancers demonstrated a significant increase in the FAD signal.

## 1. Introduction

In the US alone, 1.9 million people are estimated to be diagnosed with some form of cancer, with 600 thousand succumbing to the disease in 2021 [1]. Because of the risk and prevalence of this disease, accurate and early diagnoses are needed to improve patient outcomes. For example, one study found early detection of breast cancer improves relative survival between 27–47% [2], and another found that a one-year survival for lung cancer drops from 85% to 15% if a patient is diagnosed at stage 4 rather than stage 1, respectively [3]. 

Optical imaging has gained momentum in recent years to aid in the treatment and diagnosis of cancer. One of the most popular optical imaging modalities used is fluorescence imaging due to its low cost, potentially compact size, and real-time imaging capability. This imaging modality has been leveraged to help assess various cancers. In one study, folate hapten was conjugated with fluorescein isothiocyanate to target folate receptor alpha, causing accumulation in cancerous ovarian tissue, and aiding in imaging the intraoperative ovarian cancer cells when using a fluorescent imaging system [4]. Another example is the use of IRDye800CW to target VEGF-A, allowing for an increase in contrast of the malignant tumor and tumor margin relative to benign tissue in breast cancer patients [5]. 

Although fluorescent imaging has proven extremely useful, there are some drawbacks. For example, for many clinical imaging procedures, an exogenous contrast agent needs to be used to increase the signal of the area of interest. Although FDA-approved contrast agents are safe for patients, side effects can still occur. For example, fluorescein, an FDA-approved imaging agent, can potentially lead to low birth weights when used in pregnant women [6]. Furthermore, if a new contrast agent is developed, the approval process may be long and costly. 

One fluorescent imaging method that has started to gain popularity is the imaging of autofluorescent molecules (specifically NADH and FAD) to circumvent the need to inject fluorescent probes. In the 1950 and ’60s, NADH and FAD concentrations were found to inform the metabolism of cells and tissues [7,8]. Both NADH and FAD are electron carriers in various cellular metabolic cycles such as oxidative phosphorylation and the Krebs cycle. Because NADH and FAD function as electron donators and acceptors, respectively, measuring these metabolites can give detailed insights into the metabolic state of the tissue, for example, if there are modulations of the electron transport chain [9,10] or if tissue is oxygen/nutrient saturated or starved. In addition, by measuring the signal ratio of these two molecules, known as the (optical) redox ratio ([FAD]/([FAD] + [NADH])), the relative oxidative state of the cellular metabolism can be deduced [11]. 

Because of the autofluorescent qualities of NADH and FAD, as well as the insights that these compounds lend to tissue metabolic activity, research has been done to construct imaging systems to determine cellular metabolic footprints. One of the first systems constructed for this purpose is a low-temperature scanning system, first developed by Britton Chance and coworkers, relying on frozen tissues, shaving the tissue to a flat surface, and raster scanning the surface with a fiberoptic probe to collect NADH and FAD fluorescence [12,13]. This type of system has been used in recent years to detect mitochondrial states change in the heart [14] and kidneys [15] when injured, as well as to study the metabolic footprint of cancerous tissues [16,17,18,19,20,21,22,23,24]. In addition to the freeze-scan method developed by the Chance lab, other imaging modalities have been used to study cancer tissues in a laboratory environment, with one prominent example being two-photon microscopy (TPM). For example, TPM has been used to determine changes in breast cancer tumor microenvironment after chemotherapy [25] as well as to differentiate normal from precancerous tissue [26,27,28].

Due to the wealth of information that can be gathered about metabolism, there have been many efforts to use NADH and FAD autofluorescent imaging to aid in patient diagnostics [29]. For example, widefield redox imaging has been leveraged to aid in developing an automated high throughput system for determining patient treatment response using 3D organoid cultures [30,31], and in vivo studies have been performed to assess the metabolic state of diabetic wounds [32]. However, while widefield imaging of autofluorescence does lend insights into cancer aggressiveness and potential treatment response, the low fluorescence quantum yield of NADH and FAD and the relatively large size of widefield fluorescence imaging devices sensitive enough to image NADH and FAD limits clinical adoption. One method to overcome this limitation is to develop a handheld endoscopic imaging device to allow for easier maneuvering in the clinic. Due to the reduced distance between the surface and imaging probe, these systems can capture more fluorescent photons emitted from NADH and FAD. For example, two-photon endoscopes have been developed to extract histological information from a mouse kidney ischemia-reperfusion model in vivo [33] and to aid in the delineation of basal cell carcinomas in patients [34].

Although two-photon endoscopic devices have been developed to image NADH and FAD clinically and have been found to deliver images with high sensitivity and spatial resolution, most of these devices are relatively complex and intricate, leading to potential difficulty when used clinically. One way to overcome this complexity is to develop a simpler, more cost-effective, one photon endoscope for NADH and FAD imaging. These devices have been previously developed to aid in cervical cancer detection [35] and to measure the metabolic states of different organs [36]. As such, we have developed an easy-to-use, one photon endoscopic imaging system meant to be integrated into clinical workflow. In the current design, the photon imaging system fits into a 12-gauge breast cancer biopsy needle to allow endoscopic imaging during breast biopsy procedures.

In this paper, we build upon previous work first presented by Kanniyappan et al. [37] to develop an endoscopic detection system to measure NADH and FAD. We performed initial validation tests using phantoms containing NADH and FAD with 3% Intraliplids to determine the limits of detection of the system. Afterward, we performed tissue modulation experiments using carbonyl cyanide 4-(trifluoromethoxy)phenylhydrazone (FCCP) and rotenone + antimycin A and compared the metabolically modulated results to those found in the literature. Afterward, we measured the quantity of NADH and FAD in malignant tumor tissue ex vivo, as well as investigated the effect of photodynamic therapy (PDT) on the quantity of NADH and FAD in malignancies. We have found that our system can detect NADH and FAD in tissue, as well as detect metabolic changes in the cancer environment after treatment.

## 2. Methods

### 2.1. Construction of the Endoscopic Imaging System

Before the endoscope was developed, NADH and FAD absorption and fluorescence spectra were obtained using a plate reader (Synergy Neo2, BioTek, Winooski, VT, USA), with the NADH and FAD being excited with 375 nm and 473 nm (bandwidth 10 nm) light, respectively. 

To construct the endoscopic imaging system (Figure 1), a 375 nm laser diode (L375P70MLD, Thorlabs, Newton, NJ, USA) and a 473 nm laser diode (L473P100, Thorlabs, Newton, NJ) were used to excite NADH and FAD, respectively. Each laser diode was connected to a laser diode current controller (LDC202C, Thorlabs, Newton, NJ, USA). A dichroic short pass mirror (T450SPXRXT, Chroma Technology Corporation, Bellows Falls, VT) was placed at a 45° angle with respect to the optical axis of the 473 nm light to transmit the 375 nm light and, reflect the 473 nm light and couple the lasers lights to the excitation fiber bundle. The excitation fiber bundle was made up of polyimide buffer with a 400 µm diameter and a numerical aperture of 0.22 (Spectraconn, Rockaway, NJ, USA). To couple the 375 nm laser light to the excitation fiber, a collimating lens (f = 25 mm) and a focusing lens (f = 75 mm) were used. Similarly, a collimating lens (f = 25 mm) and a focusing lens (f = 150 mm) were used to couple the 473 nm laser to the excitation fiber bundle. To reduce/eliminate cross-talk between the channels, the NADH and FAD signals were not imaged simultaneously. Rather, the 375 nm laser would be exposed on the tissue surface and a 469 nm filter would be used to capture an NADH image. The 375 nm laser would then be blocked from being exposed, the 473 nm laser would be exposed, and the 520 nm filter would be put in place to capture a FAD image. To switch between these two lasers, a shutter (SH1, Thorlabs, Newton, NJ, USA) was used and controlled using a K-cube controller (KSC101, Thorlabs, Newton, NJ, USA). The emission coherent fiber bundle (Model No: 1533451, Schott North America, Inc. Southbridge, MA, USA) has a diameter of 1 mm, with each fiber having an 8.2 µm diameter, with a 13.5 k total fiber count. Autofluorescence is collected and transmitted to the detector by coherent fiber bundles when excited at the UV range [38], which will decrease the signal-to-noise ratio. By separating the excitation and emission fibers, we reduce the emission light passing through the imaging fiber, reducing the noise caused by the autofluorescence, and leading to an increase in signal-to-noise ratio. The endoscope tip and shaft had a maximal diameter of 2.7 mm so that the endoscope could fit easily inside of a 12-gauge needle, one of the most frequently used needle sizes for breast core biopsies [39,40,41]. The system was designed so that the light collected from the imaging fiber would pass through an objective lens (f = 50 mm), a filter wheel (FW102C, Thorlabs, Newton, NJ, USA), and then a tube lens (f = 75 mm). The filter wheel held a 469/35 nm filter (FF01-469/35-25, Semrock, West Henrietta, NY) to capture the NADH signal and a 520/35 nm filter (FF01-520/35-25, Semrock, West Henrietta, NY, USA) to capture the FAD signal. After passing through the tube lens, the signal was captured by a CCD (Model: Pixis1024−16 bit, Teledyne Princeton Instruments, Trenton, NJ, USA). The shutter system, filter wheel, and CCD were controlled using a custom-built software interface written in LabVIEW (Austin, TX, USA) and Matlab (Natick, MA, USA). 

### 2.2. Validation of the Endoscope Using Phantoms

To test and quantify the ability of our endoscopic system to image NADH and FAD, the endoscope was tested using liquid phantoms at room temperature (25 °C). The phantoms of NADH (N8129, Sigma-Aldrich, St. Louis, MO, USA) or FAD (F6625, Sigma-Aldrich, St. Louis, MO, USA) at various concentration of the analytes were made with Phosphate Buffer Saline (PBS [pH 7.4, 10010023, Thermo Fisher, Waltham, MA, USA]) with the addition of 3.3% Intralipid (I141 Sigma-Aldrich, St. Louis, MO, USA). The Intralipid phantom was meant to mimic the optical properties of human breast tissue, which has a reduced scattering coefficient (µ_s_’) of 15–20 cm^−1^ at wavelengths 350–500 nm [42,43,44], so the Intralipid was diluted to a final concentration of 3.3% (*v*/*v*) using PBS to produce the µ_s_’ of 18 cm^−1^ for NADH and 16 cm^−1^ for FAD [45]. Seven-different concentrations (0, 6.25, 12.5, 25, 50, 100, and 200 µM) of NADH and FAD liquid phantoms were prepared. Of the prepared liquid, 330µL was pipetted into a 96-well black bottom plate (655076, Greiner Bio-One, Kremsmünster, AT, USA) and imaged in the well. During the measurements, the tip of the endoscopic device was placed inside the liquid phantom, and the images were obtained using 300 ms and 1000 ms exposure times. After obtaining the signals, they were background-subtracted using the 0 µM reading. In the present study, we estimate the limit of detection by the protocol approved by the International Council of Harmonization [46]. Based on the sub-section “Validation of analytical procedures text and methodology”, the limit of detection (LOD) is calculated using the following formula:(1)$$\mathrm{LOD}=\frac{3.3\sigma }{S}$$
where *S* is the slope of the calibration curve, and *σ* is the signal standard deviation of the blank. 

To measure the penetration depth of the NADH and FAD channels, a solid resin phantom was developed. The resin phantom (EasyCast, Environmental Technology Inc., Fields Landing, CA, USA) was prepared by mixing the resin and hardener at a 1:1 ratio (by weight) and also mixing NADH or FAD in so the final mixture would have a concentration of 100 µM of either NADH or FAD. The mixture was then stirred for 10 min and allowed to cure in a low-pressure environment for 24 h to harden and remove any air bubbles. The thickness of the resin phantom was 3 mm. The endoscope was then used to image the resin phantom. Afterward, a mixture of PBS and Intralipid, with a final concentration of 3.3% (*v*/*v*), was added to the top of the resin phantom and imaged using a 1300 nm optical coherence tomography (OCT) system (Telesto, Thorlabs, Newton, NJ, USA) to confirm the thickness of the liquid layer, and then imaged with the endoscope. This process was repeated until a 1 mm and 4 mm depth was reached for the NADH and FAD phantom, respectively. 

To further validate the endoscope, we compared the endoscopic imaging system to a redox scanner previously developed by the Britton Chance lab [12,18,47,48]. In this publication, we will refer to this system as the Chance redox scanner or Chance scanner. To compare these two systems, a series of standards were measured with both systems. NADH and FAD phantom matrices were made with concentrations of 3.9, 7.8, 15.6, 31.2, 62.5, 125, and 250 µM, being diluted with a mixture of PBS + Intralipid (3.3% *v*/*v*). The details for phantom matrix preparation and the Chance redox scanner instrumentation at the University of Pennsylvania can be found in previous publications [12,19]. The phantom matrices were snap-frozen and milled flat before scanning/imaging with both the Chance redox scanner and the endoscope. Similar to the operation of the Chance redox scanner, the endoscope was positioned 80 µm above the surface of the phantom using a precision motor equipped with the Chance redox scanner. The endoscope was set to a power of 5 mW for both the 375 nm and 473 nm lasers, with the exposure time being set at 200 ms.

### 2.3. Validation and Application of Endoscope Ex Vivo

For all ex vivo experiments, we collected five images per individual point measured, the exposure time was kept at 1000 ms, and the power was set at 10 mW for the 375 nm and 473 nm laser, respectively. In addition, when imaging tissue with the endoscopic device, the endoscope was placed in direct contact with the tissue surface, eliminating any effects due to imaging distance.

#### 2.3.1. Ex Vivo Validation by Metabolic Modulation

To ensure that the signals that were being measured were from the autofluorescent molecules NADH and FAD, we performed tissue modulation experiments to determine if our endoscope could detect induced changes in tissue metabolism. The modulations were performed on MDA-MB-231 tumors (which are typically poorly differentiated grade III adenocarcinomas) that were implanted in female BALB/c mice (Charles River Laboratories, Wilmington, MA, USA). These tumors were chosen due to previously being characterized by the Chance scanner to have distinct bimodal distributions of NADH and FAD concentrations in the rim and core, respectively [18]. These mice were injected subcutaneously in the flank with 10^7^ cells per injection site and were left to grow until a tumor diameter of 12 ± 3 mm was reached. After this size had been reached, the mouse was euthanized, and the tumor was harvested and cultured ex vivo for the imaging analysis. 

Before the experiments were performed, three different solutions were prepped. One solution was PBS with 8 mM glucose (49163, Sigma-Aldrich, St. Louis, MO, USA). The second PBS solution contained 50 µM of FCCP (C2920, Sigma-Aldrich, St. Louis, MO, USA) and 8 mM glucose. The third PBS solution had a final concentration of 50 µM and 20 µM of rotenone (R8875, Sigma-Aldrich, St. Louis, MO) and antimycin A (A8674, Sigma-Aldrich, St. Louis, MO, USA), respectively.

After harvesting, each tumor piece was divided into two to three equal-sized pieces. The pieces were initially submerged into the 8 mM glucose solution for five minutes. After five minutes, the same pieces were removed from the 8 mM glucose solution and imaged with the endoscope. The pieces were then submerged in the FCCP solution for five minutes and imaged with the endoscope. And lastly, the tumor pieces were submerged in the rotenone + antimycin A solution for five minutes, then imaged. 

In total, three different mouse xenografts were used, with seven different tumor pieces being individually imaged.

#### 2.3.2. Ex vivo Tumor Imaging 

To measure the ability of the endoscopic system to differentiate normal tissue from malignant tumor tissue, we performed experiments measuring mouse xenografts of the human breast cancer MDA-MB-231 cell line. We measured the rim and core of the malignancies, as previous publications have demonstrated that in MDA-MB-231 tumor xenografts, the tumor core (central region) and rim (peripheral region) have distinct autofluorescent metabolic profiles [18].

Xenografts were prepared using the same method outlined in Section 2.3.1 Female BALB/c mice were injected subcutaneously with 10^7^ cells per injection site with tumors grown until a diameter of 12 ± 3 mm was reached. After the adequate tumor size was reached, the mouse was sacrificed, and the tumor was removed from the mouse (with skin detached). The tumor rim was then imaged, taking two to four unique and non-overlapping locations. The tumor was cut in half, with each half having the core imaged. The skin from the mouse quadriceps was then removed, with two unique locations being imaged on the quadricep. 

Four total tumors and mice quadriceps were used, with 22 rim, 19 core, and 18 quadricep images taken in total.

#### 2.3.3. PDT Treatment and Subsequent Metabolic Imaging

To determine how PDT exposure would affect the malignant tumor metabolic state, we performed PDT on breast cancers implanted in nude Female BALB/c mice. Similar to Section 2.3.1 and Section 2.3.2, 10^7^ MDA-MB-231 cells per injection site were injected into the flank of the mice and allowed to grow until a diameter of 12 ± 3 mm was reached. 

After the adequate tumor size was reached, benzoporphyrin derivative (BPD) (U.S. Pharmacopeia) was injected via tail vein injection at a concentration of 0.5 mg/kg [49,50,51,52]. After 24 h, the mouse was anesthetized and covered with a black cloth, except that the tumor was exposed. The BPD was then activated using a laser with a wavelength of 690 nm (ML6600, Modulight, Tampere, FI, USA) at an irradiance of 100 mW/cm^2^ for a total radiant exposure of 100 J/cm^2^_._ After PDT exposure, the tumor rim, core, and mouse quadriceps were imaged in the same method described in Section 2.3.2. Control mice were also imaged with the same method except that the tumors on the mice were not exposed to the 690 nm activation light. 

To determine the concentration of BPD in the tumor, fluorescent images were acquired using a fluorescent imaging system. The excitation laser from a 685 nm laser diode (HL6750MG, Thorlabs, Newton, NJ, USA) was collimated, with 8 mW of light being delivered to the sample. The fluorescence was then collected through an objective lens, passing through a 735 nm filter (FF01-735/28-25, Semrock, West Henrietta, NY, USA), with the remaining light passing through an imaging lens and being collected with a 12-bit CCD camera (EM-CCD, PCO Imaging, Kelheim, DE, USA) with an exposure time of 100 ms. To determine the approximate concentration, a standard curve was made with concentrations ranging from 0–1000 nM. 

Six mice were used in total with three in the treated group, and the other three in the control group. For the treated group, a total of 15 rim, 14 core, and 6 quadricep images were taken. For the control group, a total of 9 rim, 11 core, and 6 quadricep images were taken.

### 2.4. Data Processing

All graphs and plots were made using Prism software (San Diego, CA, USA). Figures were plotted using mean and standard error or using box and whiskers plots. The significance of the results was analyzed using either a one- or two-way ANOVA.

## 3. Results 

Figure 2 demonstrates the normalized absorption and emission spectra of both NADH and FAD solutions. The cyan and green lines in Figure 2A indicate the excitation wavelength used to excite NADH (375 nm) and FAD (473 nm), respectively, in the plate reader and the endoscope. The boxes in Figure 2B mark the bandwidths of the emission filters used on the endoscope. 

Figure 3 shows a simple validation of the endoscopic system using NADH and FAD phantoms outlined in Section 2.2. The curve was obtained using NADH and FAD concentrations between 0 and 200 µM. For NADH, the signal remained linear up to 200 µM, while some non-linearity can be seen for FAD. One reason for this non-linearity in the FAD channel is oversaturation, especially for the 1000 ms exposure time. When imaging FAD in 3.3% (*v*/*v*) Intralipid above 35 µM, the signal began to oversaturate the sensor at an exposure time of 1000 ms. Although saturation is the main cause of non-linearity in the FAD channel at 1000 ms, it is not the only reason for the plateauing effect. The FAD channel at 300 ms, which does not oversaturate the sensor, also demonstrates nonlinearity to a less extent compared to 1000 ms. Looking at the LOD for NADH we found that at an exposure time of 300 ms, the LOD was 47 µM, while at a 1000 ms exposure time, the LOD was 44 µM. The FAD channel has a LOD of 1.3 µM at an exposure time of 1000 ms.

Figure 4 (Data reproduced from: U. Kanniyappan, Xu, H.; Tang, Q.; Gaitan, B.; Liu, Y.; Li, L.; Chen, Y. Novel fiber optic-based needle redox imager for cancer diagnosis. Proc. SPIE 10489, Optical Biopsy XVI: Toward Real-Time Spectroscopic Imaging and Diagnosis, 104890J, Feb 2018, https://doi.org/10.1117/12.2293133) demonstrates the depth sensitivity of the endoscope when imaging NADH and FAD. Figure 4A,B show the 3 mm-thick 100 µM NADH and FAD resin phantoms, respectively, with the 3.3% (*v*/*v*) Intralipid solution of various heights being layered on top of the resin phantom. The endoscope acquired signals at the aqueous surface with the solution height defined as depth. Figure 4C shows the obtained signal normalized to the surface reading. The effective penetration depth was defined as when the signal intensity dropped to 1/e [53], which for NADH was 0.6 mm and for FAD was 3.3 mm. 

To further validate the system, we compared the endoscope to the Chance redox scanner for imaging the NADH and FAD phantom matrices. The results are shown in Figure 5. Figure 5A,B show the NADH and FAD signals obtained by the endoscope, respectively, and Figure 5C,D show the signals obtained by using the Chance scanner. To directly compare the signals obtained by the Chance scanner and the endoscopic imaging system, we plotted the signals obtained through the Chance scanner and endoscopic imager on the same plot in Figure 5E,F. The cross-comparison demonstrated good linearity, having an R^2^ of 0.98 and 0.99 for the NADH and FAD channels, respectively. 

In Figure 6, we see the acquired tissue NADH and FAD signals with certain metabolic modulations, where “no drug” represents the malignant tumor being placed into an 8 mM glucose solution with no mitochondrial modulators present in solution. When comparing the tumor pieces after being placed in the no drug solution and after being placed in the FCCP solution, we saw a 36% decrease in NADH, and a 31% increase in FAD, which led to a rise of 12% in the calculated optical redox ratio (ORR = [FAD]/([NADH] + [FAD])). When comparing the tumor pieces imaged after being in the FCCP solution to images after the pieces were placed in the rotenone and antimycin A solution, the NADH signal rose slightly by 6%, while the FAD signal decreased by 15%, leading to a drop in the ORR of 6%. Although we can see the expected trends in the data, the only significant difference is seen when looking at the difference in ORR between the no drug images and the FCCP images (*p* = 0.023, *n* = 7). 

Figure 7 demonstrates the signals collected from the malignant tumor and muscle tissue, with the tumor tissue having the core and the rim imaged and the muscle tissue having the quadriceps imaged. Figure 7 is divided into (A) the NADH signals measured, (B) the FAD signals measured, and (C) the calculated ORR. The core of the tumor was found to have a 30% lower NADH signal, a 55% higher FAD signal, and a 29% higher ORR value than those from the tumor rim. The differences seen between the core and the rim are significant (*p* ≤ 0.007) except when comparing the rim and core NADH (*p* = 0.077). The muscle tissue demonstrated a significantly higher NADH, lower FAD, and lower ORR than those values from both the tumor rim/core (*p* ≤ 0.0001).

Figure 8 shows the endoscopic signal obtained from malignant tumor tissue that has been injected with BPD and treated with 690 nm light at an irradiance and radiant exposure of 100 mW/cm^2^ and 100 J/cm^2^, respectively. The control mice were injected with BPD, but no exposure to light was performed. In addition, for both the treated and control groups, the muscle tissue was not exposed to the 690 nm laser light. Figure 8A shows an example of a mouse being irradiated with 690 nm light, while Figure 8B (upper) demonstrates a representative color image of the tumor pieces that were imaged using the endoscope. Figure 8B (lower) is a pseudo-colored fluorescent image looking at the accumulation of BPD in the tumor, with the concentration of BPD in the tumor being calculated at 830 ± 120 nM. Figure 8C demonstrates that after PDT exposure the rim demonstrated a 39% increase in NADH signal (*p* = 0.015), while the core saw a non-significant signal increase of 46% (*p* = 0.085). In Figure 8D, the FAD signal on the rim demonstrated a non-significant drop of 17% (*p* = 0.32), while the core saw a significant 25% signal increase (*p* = 0.0095) after PDT activation. Figure 8E shows that the calculated ORR in the rim dropped after PDT treatment by 23% (*p* = 0.0008), while the core demonstrated a non-significant drop in ORR of 8% (*p* = 0.20) after PDT activation. Moreover, when comparing the control and treated quadricep tissues, no significant differences in redox indices were seen between the two groups. 

## 4. Discussion

### 4.1. Device Validation with Phantoms 

The initial validation of the endoscopic device demonstrates that we can detect NADH and FAD present in a scattering medium similar to that of tissue, along with penetration depths between 600 and 3000 µm depending on the excitation channel. This is likely a slight overestimation in penetration depth since we only took into account scattering, and at wavelengths in the UV/blue range, blood absorption is high, which can further limit the penetration depth [44,54]. The NADH channel was found to have a LOD of 44 µM, and the FAD channel was found to have a LOD of 1.3 µM at an exposure time of 1000 ms. The higher LOD (lower sensitivity) for the NADH channel is likely due to the endoscope using an excitation wavelength of 375 nm while NADH has a peak excitation wavelength of 335 nm, as seen in Figure 2A. Although this excitation wavelength does allow us to image NADH, due to the off-peak excitation, the generated fluorescence is potentially reduced by 67% when comparing the absorptions at 335 nm and 375 nm. Although exciting at 335 nm is ideal, the reason an excitation of 375 nm was chosen was due to the limitations of UV lasers. The UV lasers that can excite at wavelengths closer to 335 nm have a larger footprint when it comes to laser heads and controller boxes. For example, a Duetto 349 UV laser head has a surface area (length x width) of 363 cm^2^, while the 375 nm laser we used had a surface area of 45 cm^2^ (this includes the temperature-controlled mount) which is an 88% reduction in size. In addition, exciting at a wavelength closer to NADH’s peak absorption cross-section of 335 nm would excite other autofluorescent molecules such as lipopigments, collagen, and elastin. These other autofluorescent molecules have emission spectra that overlap with NADH, increasing the amount of cross-talk, and leading to a decrease in signal contrast [55]. With a LOD of 44 µM, the NADH signal should still differentiate normal tissue from malignant tumor tissue because previous work has demonstrated that in normal tissue, NADH has an average nominal concentration of 84 µM, while breast tumor tissue has an average concentration of 217 µM [20]. The FAD’s LOD is also sensitive enough to differentiate between normal tissue and malignant tumor tissue, with a LOD of 1.3 µM, whereas patient samples were found to have an average FAD concentration of 83 µM and 308 µM for normal tissue and malignant tissue, respectively [20]. 

Another issue that needs to be addressed is FAD’s signal non-linearity (Figure 3B). This is likely due to the inherent quenching of FAD, leading to a decrease in fluorescence as concentration gets higher in the solution. The reason for this quenching may be due to the photoreduction of oxidized flavoquinone, with the attached adenine acting as a reducing agent [56,57]. In addition, at a high enough concentration and optical density, fluorophores can absorb the excitation light, reducing the amount of light that penetrates the sample. This means that at higher concentrations the excitation light penetrates less deeply into tissue leading to a non-linear correlation between concentration and generated fluorescence [58,59]. 

To further validate the endoscopic system, we compared NADH and FAD phantoms to the Chance scanner. As mentioned in Section 1, the Chance scanner is an imaging system used specifically to image NADH and FAD in tissue by freezing tissue, milling it flat, and raster scanning the surface with a fiberoptic probe. This method of imaging NADH and FAD has become a standard and well-accepted imaging method, being used in many studies to obtain information about the NADH and FAD distribution [12,13,14,15,16,17,18,19,20,21,22]. The NADH and FAD phantoms demonstrated good linearity when comparing the signals between the endoscope and the Chance scanner, which can be seen in Figure 5E,F. The NADH signals from both systems demonstrate the same linear pattern, while the FAD signals from both systems demonstrated a drop in linear response after 16 µM. An aspect that must be noted is that the NADH and FAD signals obtained from the endoscope during this comparison were higher than the signals obtained under normal conditions, to the point where we had to use a smaller laser power and CCD exposure time to avoid potential saturation with the sensor, with this being especially true for the NADH channel. The main reason for this is that the endoscope imaged the NADH and FAD phantoms under the same conditions as the Chance scanner, which involves imaging under the liquid nitrogen (−196 °C). At such a low temperature, NADH and FAD signals may increase up to 10-fold [48]

### 4.2. Ex Vivo Tissue Measurements

To ensure that the endoscope could image NADH and FAD signals, tissue modulations were performed using FCCP and rotenone/antimycin A. The FCCP is a potent mitochondrial oxidative phosphorylation uncoupler that results in exhaustion of NADH and increases of FAD and ORR [11,60,61]. On the opposite end, rotenone and antimycin A inhibit complex I and III, respectively, typically leading to a rise in NADH concentration, a fall in FAD concentration, and a fall in the ORR [11,60,61,62]. Although the trends we see do align with what has been previously published, it must be noted that these changes are not significant, except for the ORR signals when transitioning between going from the glucose solution to the FCCP. 

The malignant tumor core and rim were imaged because MDA-MB-231 tumor cores have been found to have a lower NADH, higher FAD, and a higher ORR signal compared to the rim [18]. Our results were largely consistent with this previous study. The exact reason for this core-rim difference is not yet known, but there is some evidence that the tumor core is less perfused, being exposed to less oxygen than the rim and potentially having damaged blood vessels [18,21]. Previous work has described five metabolic states that can be determined through the measurement of NAD, FAD, and the calculated ORR. For example, state 2 can be identified when the NADH concentration is low, and the FAD concentration is high (a high redox ratio), which means that the cells are likely being starved of necessary substrates. State 4, on the other hand, corresponds to a high NADH and low FAD concentration (a low redox ratio), meaning that the cell is in a resting state and has plenty of nutrients and oxygen [63,64]. The malignant cells in the core, due to being poorly perfused and having (potentially) fewer substrates, could be in state 2, leading to a high FAD, low NADH, and high ORR. Furthermore, hypoxia-induced limited perfusion is likely to enhance the Warburg effect in cancer cells and decrease the mitochondrial activity generating NADH for oxidative phosphorylation. On the other hand, the rim is likely supplied with more nutrients and oxygen than the core, leading to a larger NADH, lower FAD, and lower ORR, which was seen in previous studies when looking at aggressive cancers [18,21]. In addition, prior studies have demonstrated that muscle tissue is much more reduced (lower ORR) than most other tissue [36,47,65,66], which is backed up by the results obtained using our endoscope. The muscle NADH signal was larger, and the FAD signal was lower than that of both the tumor core and rim, which led to an overall lower ORR in muscle than in the malignant tissue.

In this study, we also investigated the NADH, FAD, and ORR changes that occur after PDT treatment. After the PDT excitation, we saw an increase in NADH signal from the tumor rim and core, although the change to the core was non-significant. In addition, the FAD channel demonstrated a non-significant drop in FAD in the rim, while the core showed a significant 25% rise in signal. The increase in NADH can be explained through reactive molecular species (RMS) production into a more reactive form of oxygen (singlet oxygen, superoxides, etc.). This oxygen consumption leads to a more oxygen-deprived environment, decreasing oxidative phosphorylation and increasing glycolysis to create ATP, as oxygen is needed to drive the electron transport chain, acting as an electron acceptor. As the cellular environment goes through less oxidative phosphorylation, NADH is no longer being converted to NAD+ by complex I, leading to an accumulation of NADH. The reactive oxygen species originated from exogenously administered H_2_O_2_ have been found to cause an increase in the intracellular FAD concentration [10], and the intracellular level of endogenous reactive oxygen species highly correlates with the FAD level in a breast cancer cell culture [67]. In the tumor rim, the higher amount of perfusion could lead to RMS being transported at a faster rate, leading to very little change in FAD concentration due to the lack of accumulation. On the other hand, the malignant tumor core is less perfused, potentially leading to an accumulation of RMS, leading to a larger amount of FAD in the tissue. The mouse quadricep tissue demonstrated no significant difference, which falls in line with the expected results as most of the BPD is expected to accumulate in the tumor rather than the muscle tissue [68], and the mouse quadriceps were not exposed to the activation light. Therefore, the results were expected to be the same in the control and experimental groups.

A point that needs to be addressed is the potential damage the imaging system can cause. The laser irradiance we used was 4.5 × 10^−5^ mW/µm^2^ if one looks at only the point that the excitation light intersects the tissue. While lasers can induce tissue damage, Alam et al. found that two-photon microscopes with an average power of 5.5 mW do not cause mitochondrial damage [69], with two-photon microscopes having cross-section sizes of 1–2.5 µm^2^ [70]. Using this information, we estimate that irradiances between 22–5.5 mW/µm^2^ pose little harm to the tissue mitochondria. 

### 4.3. Limitations and Future Work 

Although the system can image NADH and FAD in tissue, there are some limitations and potential future optimizations that can be performed. One limitation of the system is the low signal obtained from the NADH channel. As UV laser technology continues to advance, a smaller and more cost-effective UV laser system may be available for a future iteration of the endoscopic system, or potentially, a UV-LED may be used, although the endoscope design would need to be altered to accommodate the use of an LED rather than a laser since the lights from an LED are more difficult to be coupled into fibers. 

Other considerations that need to be taken into account when measuring the fluorescence properties of NADH and FAD are microenvironmental parameters, such as pH and temperature [71,72]. Our liquid phantom study was performed under a controlled environment, where the room temperature was at 25 °C, and the sample solvent was at pH 7.4. The tissue study was carried out at room temperature. Future studies should look to understand how the pH of tumors can impact NADH and FAD fluorescence.

Future work with the endoscopic device will involve not just comparing malignant tissue to mouse benign quadricep tissue but imaging cancers of varying levels of aggressiveness. This can include the testing and comparison of the very aggressive MDA-MB-231 cell line to less metastatic cell lines such as MCF-7 [18]. These comparisons would allow us to determine if the endoscope can not only differentiate between normal and malignant tissue but also varying levels of aggressiveness levels. Eventually, the endoscopic device may be incorporated into a biopsy needle to study clinical tumors.

Furthermore, longer-term studies need to be performed to test the ability of the endoscope to image and inform longer-term treatment response to PDT, ensuring that the treatment is taking effect. In addition, an updated endoscope will be developed to not only image NADH and FAD but also to image/activate PDT through the addition of a separate laser system specifically to excite a photosensitizer of choice because optical fibers are a common way to excite photosensitizers in the clinic. In addition, it would allow for the simultaneous imaging of a photosensitizer, giving the clinician the ability to localize PDT distribution in tissue, activate, and determine treatment efficacy using a single system.

## 5. Conclusion

In this study, we developed an endoscopic imaging system to image NADH and FAD in tissues. We found that the endoscopic system had a LOD of 44 µM and 1.3 µM for the NADH and FAD channels, respectively. Comparisons to the Chance redox scanner demonstrated a high degree of linearity when correlating the measurements between the two systems. In addition, our metabolic modulation results and the MDA-MB-231 tumor measurements align with previously published results. We also found that the endoscope could measure the metabolic impacts of PDT activation on the MDA-MB-231 malignant tumors. There was an increase of NADH and a non-significant decrease in FAD in the rim, as well as a non-significant decrease of NADH and an increase of FAD in the core. Future work will focus on increasing the sensitivity of the NADH channel, imaging the change in NADH and FAD for longer periods after PDT treatments to investigate how a malignant tumor responds metabolically to treatment in the long term, and integrating a PDT imaging/activation channel to the newly developed endoscope. We may also incorporate the current endoscope into a biopsy needle and test its capabilities with human tissue samples to help verify its potential integration into clinic use.

## Figures and Tables

**Figure 1 metabolites-12-01097-f001:**
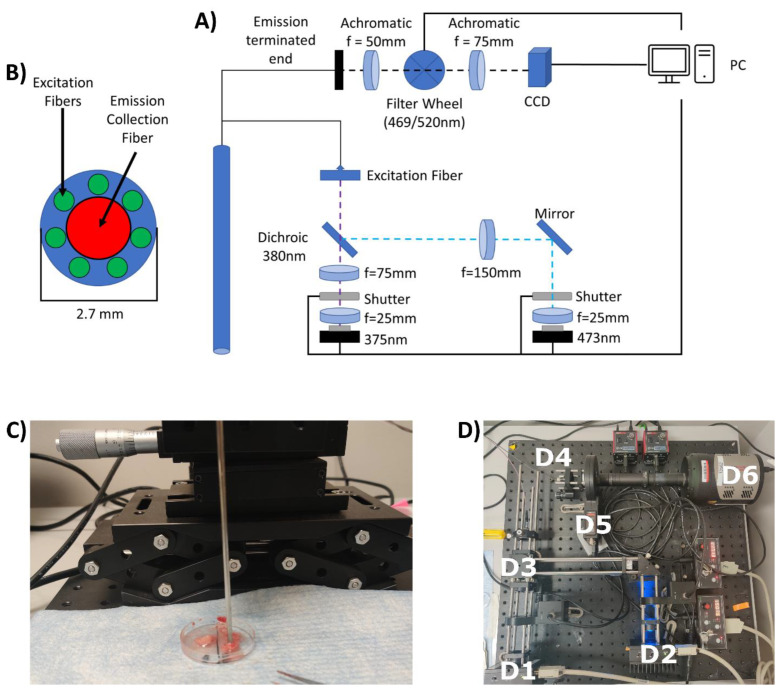
Endoscope schematics. (**A**) schematic of the developed endoscope, (**B**) a zoomed-in version of the endoscope tip, (**C**) a picture of the endoscope shaft, and (**D**) an image of the inside of the internal components of the endoscope system with the components being denoted by the following labels: (**D1**) 375 nm laser, (**D2**) 473 nm laser, (**D3**) excitation fiber bundle, (**D4**) terminal end of the imaging fiber bundle, (**D5**) filter wheel, and (**D6**) CCD.

**Figure 2 metabolites-12-01097-f002:**
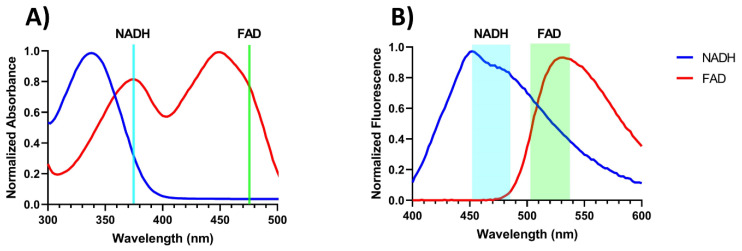
Absorption and emission spectra of NADH and FAD solutions. (**A**) Normalized absorption spectrum of NADH (blue) and FAD (red). The cyan and green lines represent the excitation wavelength used to excite the NADH and FAD: (**B**) Normalized emission spectrum NADH (blue) and FAD (red), with the cyan and green boxes representing the bandpass window of the endoscope for NADH and FAD, respectively.

**Figure 3 metabolites-12-01097-f003:**
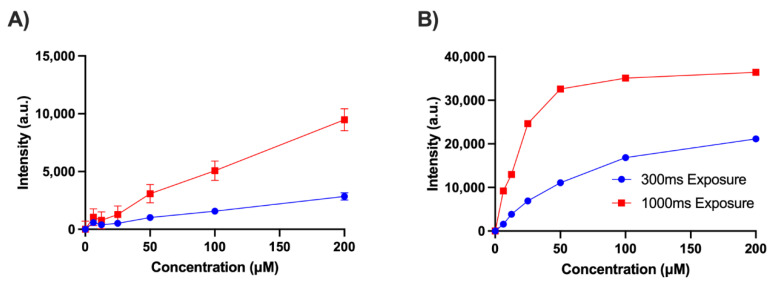
(**A**) NADH and (**B**) FAD standard curves with a concentration range between 0–200 µM with 3.3% Intralipid at 300 ms (blue) and 1000 ms (red) exposure times. The signal was background subtracted using the 0 µM readings.

**Figure 4 metabolites-12-01097-f004:**
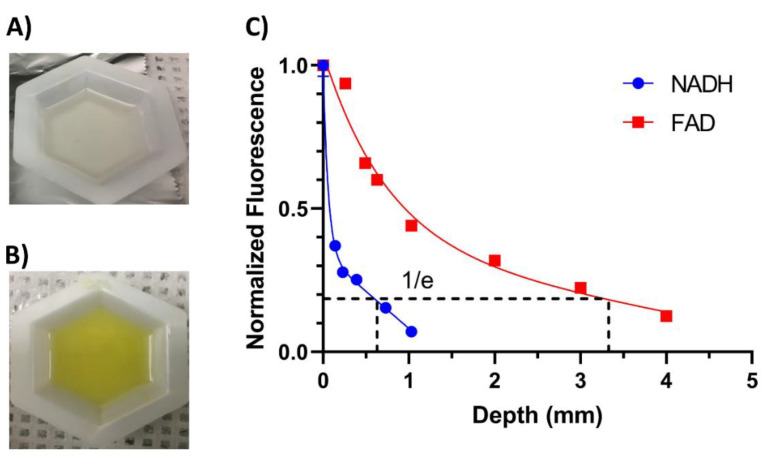
Depth sensitivity of the endoscopic system. Representative images of the 100 µM (**A**) NADH and (**B**) FAD resin phantoms: (**C**) Depth sensitivity of the NADH and FAD channel in liquid phantoms, with the Intralipid height when the fluorescence intensity dropping to 1/e of that measured at the resin phantom surface defined as the effective penetration depth.

**Figure 5 metabolites-12-01097-f005:**
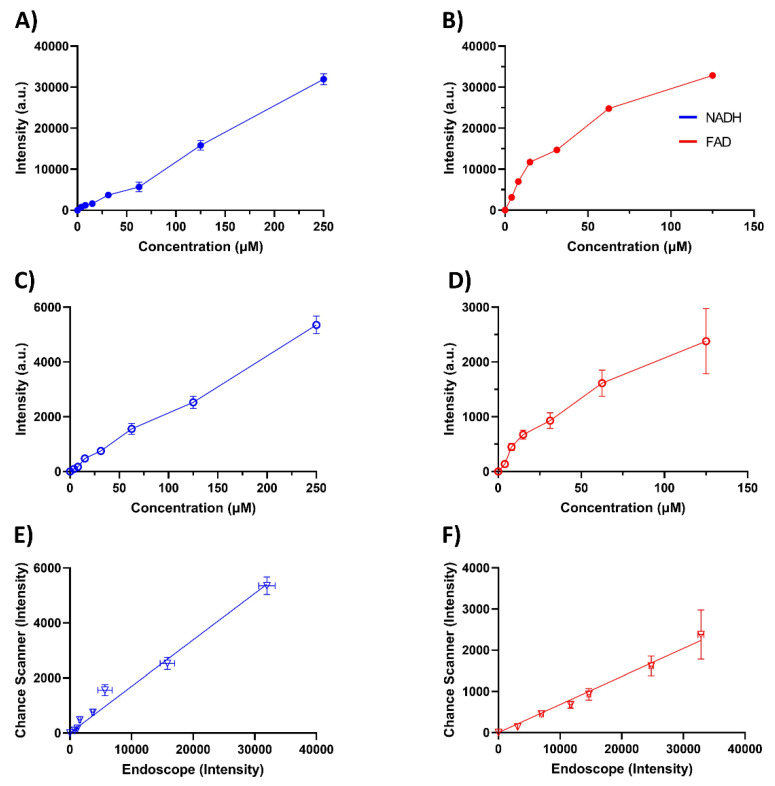
Endoscope and Chance scanner comparison using the NADH and FAD phantom matrices. (**A**) NADH and (**B**) FAD signals of the endoscopic imaging systems are displayed: (**C**) NADH and (**D**) FAD signals were obtained from the same phantom standards using the Chance scanner: (**E**) and (**F**) are the cross-comparisons between the endoscope and the Chance scanner. All blue lines denote NADH plots and red lines denote FAD plots. In addition, closed circles (●) denote signals obtained using the endoscope and open circles (◯) denote signals obtained using the Chance scanner. Triangles (∇) represent the cross-comparison plots.

**Figure 6 metabolites-12-01097-f006:**
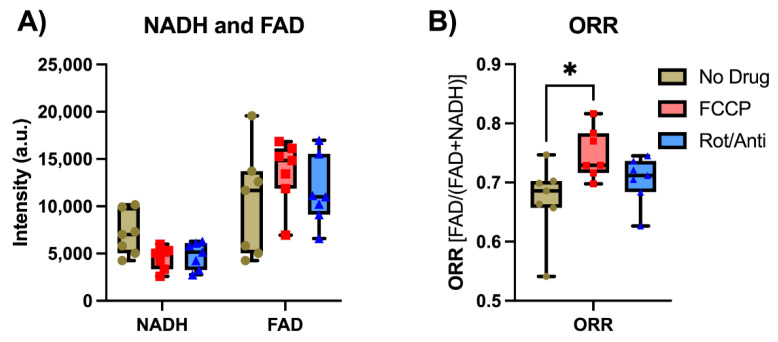
Box plots of endoscope data from tumor pieces under metabolic modulation ex vivo. (**A**) Modulation data for the NADH and FAD channels: (**B**) The calculated ORR. No drug is when the malignant tumor pieces were imaged after being placed in 8 mM glucose. FCCP is when endoscope images were taken after tumor pieces were submerged in a 50 µM FCCP/8 mM glucose solution. Rot/Anti represents when the pieces were imaged after being submerged in a solution with 50 µM rotenone, 20 µM antimycin A, and 8 mM glucose solution. All tumor pieces (*n* = 7) were imaged after being in their respective solutions for five minutes. (* = [0.05 ≥ *p* > 0.01]).

**Figure 7 metabolites-12-01097-f007:**
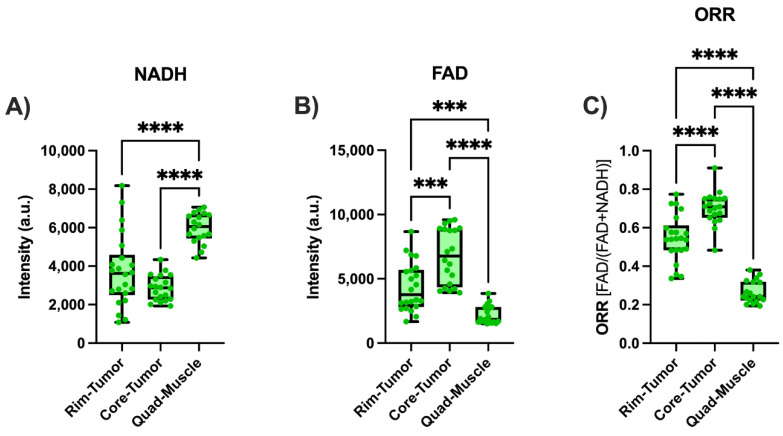
Redox indices in tumor rim and core and muscle. (**A**) NADH, (**B**) FAD, and (**C**) ORR readings for MDA-MB-231 malignant tumors and mouse quadricep tissue. (*** = [0.001 ≥ *p* > 0.0001], **** = [0.0001 ≥ *p*]).

**Figure 8 metabolites-12-01097-f008:**
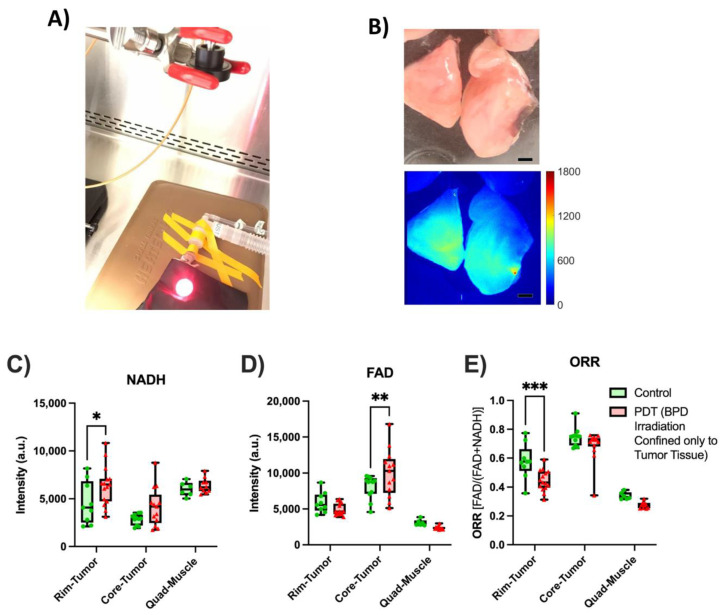
PDT treatment of malignant mouse tumor and subsequent endoscopic imaging. (**A**) An MDA-MB-231 tumor being exposed to the activation laser light after being injected with BPD: (**B**) A color image of a subdivided tumor (upper) and the BPD fluorescence (lower) in the tumor, with the color bar representing the concentration of BPD in nM. The black scale bar is equal to 1.5 mm: (**C**) NADH (**D**) FAD and (**E**) ORR readings for tumors that have been exposed to the 690 nm laser light for 100 J/cm^2^ (red), except for the muscle tissue which was not exposed. The control mice (green) were injected with BPD, but not exposed to the 690 nm laser light, and the muscle tissue was not exposed to the 690 nm laser light. (* = [0.05 ≥ *p* > 0.01], ** = [0.01 ≥ *p* > 0.001], *** = [0.001 ≥ *p* > 0.0001]).

## Data Availability

All data generated or analyzed during this study are included in this published article.

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
