# Peer review of "Development of an Endoscopic Auto-Fluorescent Sensing Device to Aid in the Detection of Breast Cancer and Inform Photodynamic Therapy"

_metabolites, 2022, doi:10.3390/metabo12111097_

Round 1

Reviewer 1 Report

no comment

Author Response

Thanks for your kind help.

Reviewer 2 Report

This manuscript “Development of an endoscopic auto-fluorescent sensing device to aid in the detection of breast cancer and inform photodynamic therapy” by Brandon Gaitan describes endoscopic device for nicotinamide adenine dinucleotide (NADH) and flavin adenine dinucleotide (FAD) in tissue. It was applied to malignant tumors. There are several major concerns as below.

-          In the first place, it is doubtful that measurement of the same tissue can be reproduced. It is not clear whether the results do not change depending on the distance from the tissue to the device.

-          Similarly, it is not only whether the results are tumor or not, but also whether the results will change depending on environmental factors such as temperature and pH. Discussion on this point is also needed.

-          Significant differences in each figure do not seem plausible. The data are scattered and do not appear to be significantly different.

Author Response

This manuscript “Development of an endoscopic auto-fluorescent sensing device to aid in the detection of breast cancer and inform photodynamic therapy” by Brandon Gaitan describes endoscopic device for nicotinamide adenine dinucleotide (NADH) and flavin adenine dinucleotide (FAD) in tissue. It was applied to malignant tumors. There are several major concerns as below.

  1. In the first place, it is doubtful that measurement of the same tissue can be reproduced. It is not clear whether the results do not change depending on the distance from the tissue to the device.

Response: There is no variation in the height because the probe is placed directly on the tissue surface. A sentence was added to specify that tissue contact was made to image the tissue on lines 217-219. In addition each individual location was measured five times to ensure consistency, with each tissue section being measured in 1-4 non-overlapping locations, as mentioned in lines 210 and 252 in the text. To ensure statistical significance, minimum of 3 different mice tumors where imaged per condition. 

  1. Similarly, it is not only whether the results are tumor or not, but also whether the results will change depending on environmental factors such as temperature and pH. Discussion on this point is also needed.

Response: This following text was added to the last section of the paper: 
Another consideration that needs to be taken into account when measuring NADH and FAD values are environmental factors that can affect the tissue microenvironment. Some examples include but are not limited to, pH and temperature. For example, FAD does not show significant changes in fluorescence lifetimes between a pH of 5 and 9 in an aqueous solution, but when imaging FAD in HeLa cells, an increase in pH led to a decrease in fluorescence lifetime
[1]. In another study, it was found that in a cell-free environment, NADH fluorescence intensity decreases as temperature and pH increase [2]. While our tissue study was performed under room temperature in the lab, future studies should look to monitor these environmental factors to get a greater understanding of how they can impact NADH and FAD fluorescence.

  1. Significant differences in each figure do not seem plausible. The data are scattered and do not appear to be significantly different.

        Response: All ex-vivo data collected was performed on at least 3 different tissue samples to establish statistical significance, and with 1-3 images taken per specified tissue section, as stated in the text. For figure 6, a two-way ANOVA was used due to 3 different conditions being compared (No Drug, FCCP being added and Rotenone/Antimycin A). For figure 7 the signal that was produced by different tissue types (tumor core, tumor rim, quadricep) is what was being compared, which counts as one factor. As such a one-way ANOVA was used to compare the tissue samples. For figure 8, in addition to comparing the autofluorecent signal of different tissue types, we also compared the signal with and without PDT activation. This means that a two-factor comparison is needed, and as such a two way ANOVA was performed. The text mentions that either a one- or two-way ANOVA was performed on line 288.

[1]          M. Islam, M. Honma, T. Nakabayashi, M. Kinjo, and N. Ohta, "pH Dependence of the Fluorescence Lifetime of FAD in Solution and in Cells," International Journal of Molecular Sciences, vol. 14, no. 1, pp. 1952-1963, 2013, doi: 10.3390/ijms14011952.

[2]          T. M. Cannon et al., "Characterization of NADH fluorescence properties under one-photon excitation with respect to temperature, pH, and binding to lactate dehydrogenase," OSA Continuum, vol. 4, no. 5, p. 1610, 2021, doi: 10.1364/osac.423082.

Reviewer 3 Report

The paper by Gaitan et al describes application of endoscopic autofluorescent (AF) sensing device for the detection of breast cancer. The theme of the study is important and intresting to a wide range of readers. The paper is well written and contains rather full description of obtained Results, has informative Discussion and adequate References. In general the paper is clearly written and Conclusions are supported by the findings of the study. 

At the same time, there are a number of major flaws that must be eliminated prior to publication.

Major Issues:

1. According to Figure 2 AF analysis of FAD is performed in 510-540 nm range. However, in this range NADH contributes to the AF signal. The authors tested the proposed device on the phantoms for FAD and NADH estimations separately, but what  the authors may expect if the phantom will contain both FAD and NADH? (note that this case corresponds to real tissue). I think, results of  measurements of such phantom, or et least adequate discussion of this question, are required in order to prove that 510-540 nm range may be utilized for FAD measurements (and that these measurements are not dominated by NADH).

2. The ethical statement is missing. Without such statement the paper may not be published.

3. The description of laser radiation density on the tissue is missing. May the utilized density damage tissues in in vivo experiments?

4. Figures 6, 7, 8 presents data as bars (representing mean values) with SD and beeplots. It is more appropriate to demonstrate data as whisker-and-box plots combined with beeswarms. 

Minor issues:

1. Captions of Figures 6-8 are missing explanations of *, **, etc.

2. Bar in Figure 8B is missing value indication.

The paper may be published after correction of mentioned issues.

Author Response

The paper by Gaitan et al describes application of endoscopic autofluorescent (AF) sensing device for the detection of breast cancer. The theme of the study is important and intresting to a wide range of readers. The paper is well written and contains rather full description of obtained Results, has informative Discussion and adequate References. In general, the paper is clearly written and Conclusions are supported by the findings of the study. 

At the same time, there are a number of major flaws that must be eliminated prior to publication.

Major Issues:

  1. According to Figure 2 AF analysis of FAD is performed in 510-540 nm range. However, in this range NADH contributes to the AF signal. The authors tested the proposed device on the phantoms for FAD and NADH estimations separately, but what the authors may expect if the phantom will contain both FAD and NADH? (note that this case corresponds to real tissue). I think, results of measurements of such phantom, or et least adequate discussion of this question, are required in order to prove that 510-540 nm range may be utilized for FAD measurements (and that these measurements are not dominated by NADH).

Response: We apologize for not describing it more clearly about the optics involved in our device, which might have caused the confusion here. In our device, we have two separate excitation lasers operating at 375 nm and 473 nm for exciting NADH and FAD, respectively. Accordingly, NADH and FAD have their own bandpass emission filters with center wavelength at 469nm and 520nm, respectively. These two channels do not acquire signals simultaneously. Since we used 473 nm to excite FAD which does not excite NADH (NADH needs UV to excite), the emission we obtained through the FAD emission filter are largely free of NADH signals, if not all. Vice versa: when we acquire NADH signals, the emission channel of NADH eliminates FAD signals despite that FAD can also be excited by the UV light. We have added more descriptions about our device in section/lines 135-139. The text is below:

To reduce/eliminate cross-talk between the channels, the NADH and FAD signals were not imaged simultaneously. Rather, the 375nm laser would be exposed on the tissue surface and a 469nm filter would be used to capture an NADH image. The 375nm laser would then be blocked from being exposed, the 473nm laser would be exposed and the 520nm filter would be put in place to capture a FAD image.

  1. The ethical statement is missing. Without such statement the paper may not be published.

Response: A conflict-of-interest statement and an Institutional Review Board Statement was added on line 562-566

  1. The description of laser radiation density on the tissue is missing. May the utilized density damage tissues in in vivo experiments?

Response: We have added this information in the revised manuscript (lines 517-523):
A point that needs to be addressed is the potential damage the imaging system can cause. The laser irradiance we used was 4.5*10-5mW/µm2 if one looks at only the point that the excitation light intersects the tissue. While lasers can induce tissue damage, Alam et al. found that two-photon microscopes with an average power of 5.5mW do not cause mitochondrial damage [1], with two-photon microscopes having cross-section sizes of 1-2.5µm2 [2]. Using this information, we can estimate that irradiances between 22 – 5.4mW/µm2 pose little harm to the tissue mitochondria.

  1. Figures 6, 7, 8 presents data as bars (representing mean values) with SD and beeplots. It is more appropriate to demonstrate data as whisker-and-box plots combined with beeswarms. 

Response: All plots have been fixed.

Minor issues:

  1. Captions of Figures 6-8 are missing explanations of *, **, etc.

Response: Added in explanations in figure caption in 6-8.

  1. Bar in Figure 8B is missing value indication.

Response: Fixed in the figure. A label was added.

The paper may be published after correction of mentioned issues.

[1]          S. R. Alam, H. Wallrabe, K. G. Christopher, K. H. Siller, and A. Periasamy, "Characterization of mitochondrial dysfunction due to laser damage by 2-photon FLIM microscopy," Scientific Reports, vol. 12, no. 1, 2022, doi: 10.1038/s41598-022-15639-z.

[2]          J. Lecoq, N. Orlova, and B. F. Grewe, "Wide. Fast. Deep: Recent Advances in Multiphoton Microscopy of In Vivo Neuronal Activity," J Neurosci, vol. 39, no. 46, pp. 9042-9052, Nov 13 2019, doi: 10.1523/JNEUROSCI.1527-18.2019.

Round 2

Reviewer 2 Report

The answers to the first and third questions seem fine. However, that to the second question is not good enough.

Fluorescence lifetime is not used in this study and is not relevant, so it should not be described.

The authors should obtain their own data and discuss this point, as it has a significant impact on the results.

Author Response

Response: As suggested by the reviewer, we have removed the discussion of fluorescence lifetime. We have now provided new experimental information that the endoscope was tested using liquid phantoms at room temperature (25 °C). The liquid phantom containing NADH or FAD of was prepared in phosphate-buffered saline (PBS) solvent at pH of 7.4.

The following text has been updated in the paper:

To test and quantify the ability of our developed endoscopic system to image NADH and FAD, the endoscope was tested using liquid phantoms at room temperature (25 °C). The phantoms of NADH (N8129, Sigma-Aldrich, St. Louis, MO) or FAD (F6625, Sigma-Aldrich, St. Louis, MO) at various concentration of the analytes were made with Phosphate Buffer Saline (PBS [pH 7.4, 10010023, ThermoFisher, Waltham, MA]) with the addition of 3% Intralipid (I141 Sigma-Aldrich, St. Louis, MO).

Other considerations that need to be taken into account when measuring the fluorescence properties of NADH and FAD are microenvironmental parameters, such as pH and temperature [1][2]. Our liquid phantom study was performed under a controlled environment, where the room temperature was at 25°C, and the sample solvent was at pH 7.4. The tissue study was carried out at room temperature. Future studies should look to understand how the pH of tumors can impact NADH and FAD fluorescence.

Reviewer 3 Report

The authors clearly addressed all my arised issues and improved the manuscript readbility. I like the way how the authors presented the findings and how they described the findings in the Discussion.

The paper may be published.

Author Response

We thank the reviewer for suggesting paper may be published.